# Primary Cell Culture as a Model System for Evolutionary Molecular Physiology

**DOI:** 10.3390/ijms25147905

**Published:** 2024-07-19

**Authors:** James M. Harper

**Affiliations:** Department of Biological Sciences, Sam Houston State University, 1900 Avenue I, Huntsville, TX 77341, USA; jmharper@shsu.edu

**Keywords:** cell culture, fibroblasts, evolutionary physiology, comparative physiology

## Abstract

Primary cell culture is a powerful model system to address fundamental questions about organismal physiology at the cellular level, especially for species that are difficult, or impossible, to study under natural or semi-natural conditions. Due to their ease of use, primary fibroblast cultures are the dominant model system, but studies using both somatic and germ cells are also common. Using these models, genome evolution and phylogenetic relationships, the molecular and biochemical basis of differential longevities among species, and the physiological consequences of life history evolution have been studied in depth. With the advent of new technologies such as gene editing and the generation of induced pluripotent stem cells (iPSC), the field of molecular evolutionary physiology will continue to expand using both descriptive and experimental approaches.

## 1. Introduction

Prior to the 20th century, studies of organismal physiology were largely restricted to intact animals, but these studies were limited in scope and difficult to interpret due to the restrictions associated with using a small number of individuals of unknown age, unknown genetic background, and unknown reproductive history, amongst other factors. During the early 1900s, the pioneering work of Abbie E.C. Lathrop, which was borne out of her business as a “fancy” mouse breeder and conducted in collaboration with scientists from the University of Pennsylvania and Harvard University, helped to establish the house mouse (*Mus musculus*) as a workhorse for experimental animal research [1]. The Norway rat (*Rattus norvegicus*) then came into widespread use several years later due to the efforts of scientists at the Wistar Institute [2], and soon thereafter, the advent of tissue and cell culture techniques for the maintenance and propagation of individual cell types opened new avenues for physiological and biochemical research [3]. Initially, these studies focused on examining the structure and composition of individual tissue and cell types but soon transitioned to examining how cells and tissues behave in response to external stimuli. This led to fundamental questions about physiological function being addressed in a diverse array of animal models. A considerable strength of both tissue and cell culture models is that neither requires the maintenance of live animal colonies, which is often impractical, if not impossible, for some species due to the space and cost limitations. It also allows for populations to be sampled directly in the wild with minimal disruption, if done properly. Please see Jimenez and Harper [4] for a recent historical perspective of cell culture techniques and their application in comparative physiology. In this mini-review, I will highlight the utility of single cell type cultures to address questions in evolutionary physiology at the molecular level. Although they are a powerful experimental model system, the use of organ slices [5,6] will not be addressed here. On the other hand, 3D organoid cultures derived from induced pluripotent stem cells (iPSC) have gained prominence as a tool to address questions about the molecular mechanisms underlying major evolutionary trends because they retain natural, species-specific properties that mimic specific properties across a wide range of tissues. Please see [7] for a brief history of the development of organoid cultures. Notwithstanding that it is beyond the scope of this review, recent examples of using organoid models to address fundamental question in evolutionary biology include the influence of copy number variation in genetic loci associated with primate neurodevelopment [8,9], as well as gene regulatory networks associated with the cardiopulmonary development and the transition to a terrestrial lifestyle [10]. Another approach relies on venom gland organoids derived from snakes to elucidate the processes that have led to the evolution of venom glands from salivary glands, as well as the venom types themselves [11]. As this technology expands to include organoids derived from a diverse range of species, for example, agricultural and companion animals, so too will the range of questions that can be addressed such as the evolution of pathophysiological mechanisms underlying zoonotic disease transmission, which is an area of growing concern [12]. 

## 2. Cell Culture Models

Although the terms are often used interchangeably, tissue culture and cell culture do not refer to the same technique. Tissue culture involves the maintenance of intact tissue explants in vitro that were collected via biopsy in order to study the response to various stimuli. Meanwhile, cell culture (typically) refers to the maintenance of a monoculture of a specific cell type (e.g., fibroblasts, hepatocytes, T-cells) under artificial conditions, although mixed cultures are sometimes used. Notably, tissue explants can be used to generate individual cell cultures, such as for the derivation of pulmonary fibroblasts [13] or epithelial cells from various organs such as the gastrointestinal tract [14]. However, it is more common to digest tissue fragments with proteolytic enzymes after mincing to “free” cells from the initial biopsy for the generation of individual cell cultures. 

In many cases, primary cell cultures eventually give way to immortalized cell lines. Immortalization can be the result of spontaneous processes [15,16] that are the consequence of the stresses associated with being moved to an artificial environment [17], or they may be immortalized via experimenter-induced processes such as mutagenesis or cell transfection [18]. Once immortalized, the cells are referred to as cell lines. Oftentimes, these immortalized cell lines also undergo transformation to become cancer-like and lose many of the properties typically associated with normal cells such as contact inhibition. Undisputedly, the ability to indefinitely propagate an individual cell line in culture is useful since it only needs to be derived once. However, to what degree these immortalized/transformed cell lines behave relative to a normal cell is often unknown and/or can vary considerably from one cell line to the next. As an example, the proteome of exosomes analyzed from Jurkat cells, an immortalized human T-cell line, does not overlap extensively with those obtained from primary cultures of T-cells [19]. Moreover, the lack of functional overlap is often true among individual cell lines of the same cell type. For example, the degree to which various glial cell lines undergo myelination varies considerably from one line to the next in both murine and human glial cell lines [20]. More problematic, however, is the genetic heterogeneity that arises with the continuous culture of individual cell lines over time due to random genetic drifts at both the genomic [21,22] and epigenetic level [23,24,25] in addition to transposition events [26]. This means that the cell line of today can be very different from the cell line of tomorrow from both a genetic and a functional perspective, which is something that is often overlooked. 

On the other hand, primary cell cultures consist of populations of individual cells that were isolated directly from a tissue biopsy. Primary cell cultures often have a limited capacity for expansion, and individual cell lineages within the population have a finite lifespan. Although this introduces the challenge of isolating and/or generating enough cells to conduct individual experiments, it varies considerably from one cell type to the next. For example, primary hepatocytes maintained in vitro lose their replicative capacity immediately [27], while primary fibroblasts isolated from a variety of organs, but most often the skin, can be expanded to generate tens- to hundreds-of-millions of cells over time. In many cases, batches of cells can be cryopreserved and stored for later use. Nevertheless, individual cultures of propagating cells eventually become replicatively senescent with subculturing (i.e., passaging) and will eventually stop dividing due to a critical shortening of the telomeres [28]. Indeed, foundational studies of cellular senescence were conducted using primary cultures of fibroblasts from multiple tissue sources [29], but this limitation can be overcome if the cells are frozen at an early passage number.

Despite these limitations, there are several strengths associated with using primary cell culture models. First, since individual cultures of primary cells are generated from discrete tissue biopsies collected from unique donors, the confounds associated with the pseudo-replication inherent to the testing of replicates of a clonal cell line are avoided. It also allows for factors such as age, sex, reproductive history, and the like to be included as experimental factors. This is not possible with established cell lines since they are limited by the identity of the original donor. Most importantly, however, primary cells maintain the morphology typical of a given cell type, as well as the cell surface and cytosolic markers that are critical for normal cellular function, making primary cell cultures a powerful model to study how these cells would behave in vivo [30]. 

Both primary cell cultures and immortalized cell lines are well-established models for pharmacological profiling, the study of gene-specific diseases processes and the changes in cellular morphology, biochemistry, and physiology that are associated with neoplastic transformation, including the evolution of cancer drug resistance and the metastatic potential of cancer cells, amongst other processes with direct biomedical applications. Although not as extensive, cell culture models are also utilized to study basic questions in evolutionary physiology down to the molecular level. Without question, primary cultures of fibroblasts are the most heavily used because of their ubiquitous distribution throughout an organism, high replicative potential, and ease of use (Figure 1). Fibroblasts grow readily in many different media formulations and often must be selected against when trying to establish cultures of other cell types such as keratinocytes or endothelial cells [4,31,32]. Interestingly, the source organ from which a fibroblast culture is derived can significantly alter the response of individual cultures to exogenous signals and should be considered when conducting experiments [33]. Here, I focus on the use of primary cell culture models for evolutionary questions concerning each of the following three areas: (1) Gene profiling, Genome Evolution, and Phylogenetic Reconstruction; (2) Longevity Associated Gene Expression and Cellular Function; and (3) Life History Evolution (Figure 2).

## 3. Genetic Profiling, Genome Evolution, and Phylogenetic Reconstruction

Cell culture models are well suited for cytogenetic studies [34] and have been conducted using cells derived from a wide array of vertebrate species. Although fibroblasts are the dominant cell type, the nucleated lymphocytes of non-mammalian vertebrates are a ready source for cytogenetic studies [35,36], while a recent study in birds used gonad-derived cells [37]. In general, it is relatively easy to generate high quality chromosome spreads from cultured cells which are amendable to karyotyping using standard staining techniques, as well as fluorescence in situ hybridization (FISH). Oftentimes, the biggest challenge for individuals using cells from non-traditional models (i.e., mammals and birds) is the need to determine the ideal culture conditions for the propagation and maintenance of individual cell types such as those from fishes [38,39,40]. Cell culture is also a powerful technique for basic cytogenetic studies when working with rare or endangered species because it allows for the generation of large numbers of cells for study while preserving other materials for anatomical, histological, and/or functional studies [41,42]. Alternatively, if a tissue biopsy is collected using a minimally invasive technique, such as an ear punch or toe clip, the sampled individual can be returned to the population. 

Once they have been generated from individual cultures, the resultant karyotypes can be used to address different questions about genome evolution, the simplest being phylogenetic reconstruction at various taxonomic levels such as orders [43], families [44,45,46], or genera [47]. Within species, this approach can also be used to tease out phylogeographic relationships among populations [48,49]. In addition to using karyotypes, primary cell cultures are suitable for phylogenetic studies using methodologies such as satellite DNA sequence analysis [50,51] or more novel approaches such as using patterns of differential gene expression among individual cultures from various species as indicators of phylogenetic relatedness [52]. In other instances, primary cell cultures have been used to study the role of heterochromatin in karyotype reorganization in fishes [53], molecular mechanisms underlying the evolution of polyploidy in fishes [54], the variation in telomere dynamics among mammal species [55], and the biology of retrotransposons, well-known drivers of genome evolution [56]. 

## 4. Longevity Associated Gene Expression and Cellular Function

An intriguing question in the comparative biology of aging is what accounts for the dramatic difference in lifespans seen across vertebrate species despite them largely being built from the same “parts”? More specifically, the structural organization and biochemistry of skeletal muscle, for example, are the same in mice and humans, but the typical human will live 40 times (or more) longer than the typical mouse. As noted above, it is not practical, or even possible, to maintain captive populations of many different individual species to address this question. However, using primary cell cultures derived from individual species with disparate lifespans offers a simple alternative to the use of populations which have been exploited by biogerontologists for several decades. It also lends itself to studies examining which mechanism(s) may account for differential longevities within species such as single-gene mutations [57,58] or selective breeding [59]. For the sake of brevity, I will highlight some of the most significant findings from studies using primary cell cultures to study the biology of aging among vertebrates. Please see Alper et al. [60] and Jimenez and Harper [4] for the recent reviews. Moreover, within non-mammalian species, the short-term culture of erythrocytes has become a popular model system to study physiological function because large numbers of cells can be obtained from a simple blood draw, and because non-mammalian erythrocytes are nucleated and retain their mitochondria. Hence, they have high metabolic capacity and are amendable to studies of mitochondrial function, as well as telomere dynamics, an oft studied, if still contentious, area of research in biogerontology. Please see Stier et al. [61] for an overview of using erythrocytes for aging studies. 

Because invertebrate models of differential aging have indicated that long-lived individuals tend to be more resistant to the lethal effects of a variety of exogenous stressors, such as oxidizing agents, heat, or heavy metals [62], initial studies using primary fibroblast cultures asked whether cells from long-lived species were more resistant to stress than those from shorter-lived species. Although not absolute, this was generally the case in both mammals [63,64,65,66] and birds [67,68,69]. While a specific mechanism that underlies this differential stress resistance has not been identified, primary cell cultures of fibroblasts have provided hints at the evolutionary origins of these processes. For example, Dostal et al. [70] demonstrated that fibroblasts from long-lived rodents have reduced uptake of the heavy metal cadmium in conjunction with reduced levels of metal ions that are active in redox reactions compared to cells from short-lived rodents. In addition, cells from long-lived mammals significantly upregulate the expression of DNA repair genes and those involved in glucose metabolism with clear phylogenetic underpinnings [71], as well as the differential expression of antioxidant enzymes [72]. Similarly, the metabolic function of cells isolated from small, long-lived dog breeds is different from that of those isolated from large, short-lived dog breeds [73]. Furthermore, in both bird and mammal cells, increased donor longevity was associated with differential activation of extracellular signal-regulated kinase 1/2 (ERK-1/2), an important mediator of cellular stress resistance [74]. Interestingly, while not directly related to aging research, using primary cultures of fibroblasts from a variety of species to examine the evolution of toxin resistance in the context of ecotoxicology has also become common [75,76,77,78]. 

In addition to differential stress resistance, studies of primary cell cultures from species with differential lifespans have indicated that the evolution of proteome maintenance and turnover is likely to be an important contributor to the aging process. More specifically, proteins isolated from fibroblasts grown from long-lived species exhibit reduced levels of protein carbonyl when compared to those grown from short-lived species [79,80,81], while the rate of protein turnover is also reduced in long-lived species [82,83,84] in conjunction with a differential induction of autophagy [85]. There is also evidence that the remarkable longevity of the naked mole rat may be due, at least in part, to the ability of their cells to resist malignant transformation due to differences in cell behavior [86] and biochemistry [87,88,89]. Cells from the blind mole rat (*Spalax*) are similarly cancer resistance, and can even directly kill cancer cells, perhaps as an adaptation to the hypoxic conditions they face as a subterranean mammal [90]. On the other hand, the evolution of differential oxygen sensitivity does not seem to be involved with the aging process except in the case of the peculiar evolutionary history of the laboratory mouse [91].

The relationships described above are robust and have consistently held up when moving from one model to another, but they have heavily utilized fibroblasts, most often isolated from the skin. In a recent study, it was found that the degree of resilience of primary cell cultures from different tissues and tissue regions, as well as different cell types, varied in their response to various insults [92]. Although this is not a surprising outcome, it does call into the question the reliance on a single cell type to infer broad physiological relationships for complex processes such as aging. 

## 5. Life History Evolution

According to the tenets of life history theory, species tend to adopt a lifestyle that maximizes their reproductive fitness and lies along a ‘slow–fast’ axis [93]. Species on the ‘slow’ end of the axis tend to be slow growing with a late age of first reproduction in conjunction with a low mass-specific metabolic rate and a delayed onset of senescence. Conversely, species at the ‘fast’ end of the axis tend to high have mass specific metabolic rates, exhibit rapid growth, and have an early onset of senescence at the whole organism level. As with the evolution of longevity assurance mechanisms, primary cell cultures are an attractive model to determine how variation in life-histories translates to cellular function. 

Interestingly, in both mammals [94] and birds [95], there is no correlation between cellular metabolic rate and body mass across species despite their dramatic differences in mass-specific metabolic rates. On the other hand, there are marked differences in stress resistance parameters that have been in concert with the individual life-history strategies in birds [69], perhaps downstream of the individual growth rates as demonstrated in cultures of fibroblasts and myoblasts [96,97]. Differences in the mitochondrial membrane lipid composition also varied as a function of individual life history strategies [98], while a more sophisticated approach using interspecific cybrids from bovids teased out the contribution of species-specific mitochondria to cellular metabolism [99]. Meanwhile, primary adipocytes from bears have been used to study the molecular basis of cellular adaptation to the metabolic stress of hibernation [100], while primary fibroblasts from reindeer have demonstrated that circadian clock genes are absent in this species, most likely because of the abnormal photoperiods experienced by arctic species [101]. 

On the other hand, body size does correlate with a p53-mediated DNA damage response in elephants [102], and reproductive history may also be an important modulator of cellular function, at least in mammals [103]. What remains to be seen is how other physiological processes such as endoplasmic reticulum stress and the unfolded protein response at the cellular level are shaped by ecology and evolution [104], although increased ER stress sensitivity has been seen in both a reptile [105] and a mammal [65] at the slow end of the axis. 

## 6. Future Directions

Primary cell culture is a critical tool for conducting genome-to-phenome studies and are especially useful for species that are impractical to study under captive conditions such as aquatic species [106], leading to published protocols for the isolation of various cell types from both fishes [39] and marine mammals [107]. Undoubtedly, cell cultures derived from these, as well as endangered [108,109], species will continue to contribute to our understanding of evolutionary adaptations to different environments while helping to preserve individual species, but the application of “modern” techniques in cell culture and molecular biology are poised to move the use of primary cell cultures into new directions. This includes using transfection of individual cultures to generate immortalized cell lines [17], but this has the potential to change how the cells behave in an unpredictable manner [110,111]. 

### 6.1. CRISPR/Cas9 

Techniques for the genetic manipulation of primary cells have existed for many years and have provided a means to examine the effect of specific genetic manipulations on cellular function [112]. The emergence of CRISPR/Cas9 technology has promised to take this to a new level due to its relative ease of use and applicability to all types of biological material [113], with optimization being necessary for some species [114,115]. In the context of primary cell culture, CRISPR/Cas9 has been used to demonstrate the role of the mitochondrial transcription factor, TFAM, on mitochondrial function in a bovine model [116] and the evolution of the integrated stress response in vertebrates [117]. This approach has also been used for studies of evolutionary developmental biology by manipulating ancestral genes in lampreys [118] fibroblast growth factor 5 (FGF5) in sheep [119], as well as the role of the CLOCK gene in the evolution of circadian time maintenance in eukaryotes [120], although here early-stage embryos or zygotes were used rather than cell cultures. Moreover, CRISPR/Cas9 technology provides a powerful means of creating specific models to address questions in evolutionary molecular physiology. For example, the experimental evolution of differential alcohol tolerance among yeast strains, followed by CRISPR/Cas9-mediated targeting of evolved mutations, allows for the direct confirmation of the role of specific mutations in the evolved trait, as well as putative trade-offs accompanying individual mutations [121]. This same approach could be readily applied to primary cultures of various cell types exposed to varied environmental conditions, such as heat or varying nutrient availability. Because genomic data are absent for most species, especially non-traditional or exotic model organisms, the primary hindrance to the widespread use of CRISPR/Cas9 technology to address central questions in evolutionary physiology is the lack of specific genomic sequences to be targeted. Since it is a gene editing tool, the simplest analogy for the CRISPR/Cas9 system is a genetic word processor that is used to correct typos or change the text in the form of nucleotide sequences rather than characters. However, as next generation sequencing (NGS) technologies continue to become increasingly rapid and cheaper, this limitation should be rendered moot for many species of interest soon enough. 

### 6.2. Induced Pluripotent Stem Cells (iPSC)

Induced pluripotent stem cells (iPSC) were first described in 2006 [122] and refers to the generation of a pluripotent stem cell via the forced expression of key transcriptional regulators in somatic cells. While fibroblasts are the most used for the reasons cited previously, iPSCs can be generated from other somatic cells and can be differentiated to generate almost any somatic cell type, intact tissues or organs, or even whole animals via cloning [123]. Using this approach, the limitations of using a population of fibroblasts as a proxy for other cell types, or even as a proxy for fibroblasts from another tissue/organ sources, could be avoided. The list of species used to generate iPSCs is ever growing and, within mammals, includes humans and other primates, multiple rodents, multiple species of carnivores, and agriculturally important species such as pigs. Within birds, iPSCs have been generated from common domestic species such as chickens and zebra finches, but these lists also include exotic species such as the Tasmanian devil [124,125]. Examples of using this technique to address deep evolutionary questions, include the generation of an iPSC from a monotreme mammal (platypus) to dissect the role of specific factors in the evolution of pluripotency based on the differential expression of individual genes in eutherian mammals versus monotremes [126], while Hirata et al. [127] compared the transcriptome and epigenome of iPSCs from humans and chimpanzees to reveal their presumed contribution to the phenotypic divergence seen among these closely related species. Similarly, iPSC lines from horse, donkey, and mule demonstrated marked differences in cell growth, pluripotency, and patterns of differentiation in the mule iPSCs relative to the parental strains to provide material for the study of heterosis and hybrid gamete formation [128]. An iPSC line generated from a rodent with an unusual chromosome constitution involving the sex chromosomes provides a unique resource to elucidate the evolution of mammalian sex determination [129,130]. 

### 6.3. Transformed Human Cell Lines

In another approach, established human cells lines can be used to express individual gene products to ascertain their effect from an evolutionary perspective. Although not a primary cell culture model, this system is gaining momentum and has made significant contributions to understanding the evolution of complex physiological processes. Human embryonic kidney (HEK) 293 cells are one of the most widely used for these studies because they are easy to transfect, have high growth potential, and can be cultured under a variety of conditions [131]. Notable studies using the approach are the identification of key enzymes involved in carotenoid metabolism [132], the evolution of combinatorial coding of olfaction in canines [133], and the evolution of retinoids in the evolution of the vertebrate visual system [134]. In the first instance, the authors were able to confirm the role of specific enzymes in ketocarotenoid synthesis in both birds and fishes, where they play an important role in social interactions. In the second case, a subset of canine odorant receptor genes was expressed in HEK293 cells, and their response was monitored after the application of specific odorants. Finally, using bioinformatics, it was inferred that lampreys possessed enzymes that were homologous to those critical for enzymatic regeneration of 11-cis-Retinal in the vertebrate eye and were found to be functional when expressed in HEK293 cells. 

## 7. Conclusions

Primary cell culture is a useful model to address the fundamental questions in evolutionary physiology since primary cells retain a ‘normal’ phenotype that recapitulates how the cells would behave in vivo. For the study of the molecular and biochemical mechanisms underlying genome evolution, longevity assurance mechanisms, and the variability in life history traits among species and populations, primary cell culture have been especially beneficial. Nevertheless, primary cell culture has also been used to infer the mechanisms underlying the evolution of sex determination in both cartilaginous and bony fishes [43,135], the evolution of innate immune function in eels [136], variation in skeletal traits among primate species [137], glial cell evolution in mustelids [138], hypoxia tolerance in ground squirrels [139], differential gene expression due to chromosomal fusion in mammalian germ cells [140], factors involved in the evolution of the mammalian uterus [141], and the evolution of skin development [142] in fishes to name a few. Except for [42], fibroblasts were not the cell type used for these studies, which is in stark contrast to most of the studies that have been cited. Undoubtedly, this reflects the relative ease in generating primary cultures of fibroblasts due to their abundance and ubiquitous distribution in situ and their ability to tolerate a wide range of culture conditions in conjunction with their high replicative potential. However, as demonstrated in [92], limiting studies to a single cell type can mask important relationships. Even inferring what occurs in a single cell type such as fibroblasts is erroneous since the tissue or organ from which it is isolated can significantly alter the outcome [32,92]. Moreover, since the primary role of fibroblasts in vivo is the establishment and maintenance of the extracellular matrix, in addition to playing a prominent role in the wound healing response, they cannot serve as surrogates for other cell types with specialized functions (e.g., myocytes). 

## 8. Perspective and Significance

Primary cell cultures have proven their worth for those interested in studying evolutionary physiology at the molecular level and are amendable to studies of experimental evolution. In addition, the biochemical pathways that are central to organismal function, such as stress defense mechanisms, DNA repair, protein deamination and transamination, and so on are retained by primary cultures, and these cells will behave in a manner which is consistent with that seen in an intact organism. Moreover, primary cell cultures that have been established using donor individuals from different species, different populations of individual species, or even genetic variants within a species provide a powerful tool to address important question in evolutionary physiology at the molecular level. Although this review has emphasized the utility of classic two-dimensional cell culture models of individual cell types, there are limitations associated with this approach, the most significant being that it is a poor representation of the in situ condition. Individual cell types do not exist in isolation from others, nor do they exist in a monolayer that is provided with nutrients from the top down. In addition, the regular disruption of cells due to subculturing and the provision of blood serum rather than plasma is akin to repeated wounding rather than physiological stability. Two-dimensional models persist mostly due to their relative ease-of-use, a limited need for specialized equipment, and their legacy as a model system. 

Nevertheless, there is tremendous interest in developing other cell culture systems that more closely recapitulate physiological normality. The simplest of these are mixed-cell (co-culture) systems that allow either of the following: (1) direct interaction of individual cell types via physical contact in heterogenous monolayers or on a three-dimensional scaffold, or (2) indirect interaction using conditioned media, a “shared” extracellular matrix, or porous membranes (e.g., Transwell^®^). In the latter case, the interaction is limited to the exposure of one cell type to factors secreted by another other cell type [143]. The most common use for this approach is the study of the influence of tumor cells on “normal” cells in the immediate area, but it can be asked to address fundamental questions in evolutionary physiology as well, such as the role of neuro-immune interactions in the evolution of the vertebrate nervous system [144].

Hollow fiber bioreactors (HFBR) are a continuous perfusion cell culture system whereby the cell culture medium is circulated through fibers whose outsides have been seeded with cells. This allows for nutrient and gas exchange to occur across the fiber walls in both directions for the continuous and controlled oxygenation of cells and nutrient delivery using a controlled medium composition. It also allows for the delivery of circulating factors such as signaling molecules or drugs in a manner that is consistent with their delivery to individual tissues via capillaries. Finally, it is possible to coculture multiple cell types in a system that more closely resembles intact tissues [145]. While HFBR systems are in widespread use within the biotech sector, there have been no studies using them to address the fundamental questions in evolutionary molecular physiology; thus, it would be beneficial to adapt them for the examination of the physiology of various cell types, either alone or in combination, from species with significant differences in maximum lifespan, for example. Undoubtedly, the continued use of these cell culture models will continue to provide important insight for evolutionary biologists. 

## Figures and Tables

**Figure 1 ijms-25-07905-f001:**
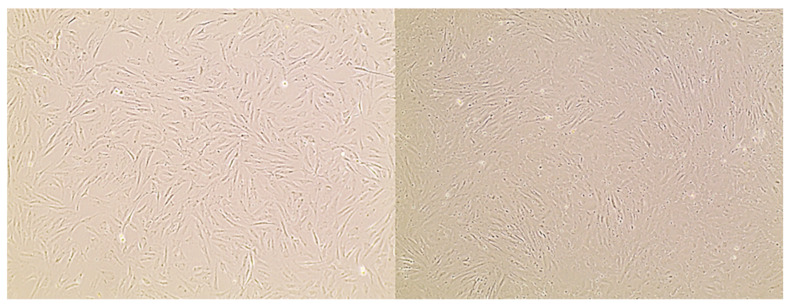
Primary cultures of fibroblasts, such as these from a turtle (**left**) and a bird (**right**) are the most used model for studies of comparative physiology. They are widely distributed in connective tissue, are easily isolated from tissue biopsies, grow readily in many different media formulations, and have a high proliferative potential. 4X magnification.

**Figure 2 ijms-25-07905-f002:**
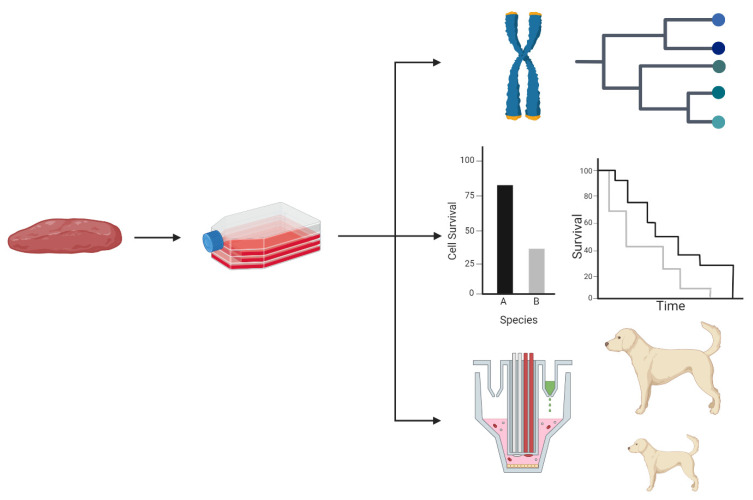
Primary cell cultures derived from tissue biopsies can be used to address fundamental questions in evolutionary biology such as phylogenetic reconstruction using chromosome spreads (**top**), the mechanistic basis for differential survival among species (**middle**), or the relationship between cellular and whole animal physiology within individual species (**bottom**).

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
