# Peer review of "Primary Cell Culture as a Model System for Evolutionary Molecular Physiology"

_ijms, 2024, doi:10.3390/ijms25147905_

Round 1
Reviewer 1 Report
Comments and Suggestions for Authors
The author performed a conceptually sound study.
James M. Harper has comprehensively reported how to utilize primary cell culture to study organismal physiology.
The review started from the introduction of cell culture models, the author reported the pros and cons of these tools. The author tried to reveal the availability, advantages, future evolutions, perspectives and limitations of primary cell culture to study evolutionary physiology.
Primary cell culture provides a powerful tool for the study of the molecular and biochemical mechanisms underlying genome evolution, longevity assurance mechanisms, and the variability in life history traits among species and populations. I think it is relevant and interesting.
The “Conclusions” and “Future directions” are presented and discussed adequately and consistent with the arguments and evidence/references presented.
The paper is well written, clear, and easy to ready.
I have no comments that would improve the manuscript. Thus, I recommend acceptance of the manuscript in its present form.
Author Response
Thank you for feedback.
The reviewer has no specific comments/suggestions to address.
Reviewer 2 Report
Comments and Suggestions for Authors
The section Future Directions contains essential and interesting information that should be presented before conclusions. It should be separated into at least two sections, one about gene editing and the second about iPS and organoids.
The gene editing and CRISPR/Cas9 technology section should include more recent studies and explain how this technology contributes to creating models for the study of molecular physiology.
A section on two-dimensional cell culture systems could discuss other cell types, such as mesenchymal cells, in more detail in addition to fibroblast cultures. Coculture systems such as transwells should also be mentioned.
Bioreactor applications, such as the hollow fiber type of bioreactor that offers an environment close to physiological, should also be mentioned.
A separate section should present specific examples of induced pluripotent stem cells (iPSCs) and their use in evolutionary studies. The use of iPSCs' derived organoids is another significant field that would require a separate section. Organoid applications in molecular physiology studies should be compared with those of two-dimensional cell cultures and bioreactors.
Author Response
Thank you for your careful reading of the manuscript and your thoughtful comments and suggestions. I have addressed each in turn and I feel they have improved the quality of the manuscript substantially. Please see my response to each comment below:
Comment 1: The section Future Directions contains essential and interesting information that should be presented before conclusions. It should be separated into at least two sections, one about gene editing and the second about iPS and organoids.
Comment 2: The gene editing and CRISPR/Cas9 technology section should include more recent studies and explain how this technology contributes to creating models for the study of molecular physiology.
Response to Comments 1 and 2: The future directions section has been moved and is presented before the conclusions section. It has been separated into individual sections as recommended as has been expanded to include additional, and more recent studies using CRISPR/Cas9. An example (Lines 277-284) has been added to illustrate how this approach can be used to create models for the study of evolutionary molecular physiology.
Comment 3: A section on two-dimensional cell culture systems could discuss other cell types, such as mesenchymal cells, in more detail in addition to fibroblast cultures. Coculture systems such as transwells should also be mentioned.
Comment 4: Bioreactor applications, such as the hollow fiber type of bioreactor that offers an environment close to physiological, should also be mentioned.
Response to Comments 3 and 4: A brief discussion of co-coculture model systems and HFBRs has been added to the Perspective and Significance section (Lines 375 – 406)
Comment 5: A separate section should present specific examples of induced pluripotent stem cells (iPSCs) and their use in evolutionary studies. The use of iPSCs' derived organoids is another significant field that would require a separate section. Organoid applications in molecular physiology studies should be compared with those of two-dimensional cell cultures and bioreactors.
Response to Comment 5: The section on iPSCs has been expanded to include additional references highlighting the utility of this model for studies of evolutionary physiology (Lines 312 – 320).
A discussion of iPSCs to derive organoids, and organoid applications in general, for evolutionary physiology has not been included. Although this is a powerful model system, it would require an extensive introduction and could be the focus of a standalone review. Since the current manuscript is meant to focus on primary cell culture as a model system the exclusion of organoids was addressed directly early in the original manuscript (Lines 41 – 44).
Round 2
Reviewer 2 Report
Comments and Suggestions for Authors
The manuscript is significantly improved based on the reviewer's comments. Most comments have been adequately addressed. The reviewer believes that a brief discussion on organoid applications would be valuable. The introduction could briefly mention the role and potential of organoids in Evolutionary molecular physiology to set the stage for a more detailed discussion in future works. This way, the significance of organoids would be acknowledged while the manuscript's primary focus would be maintained.
Author Response
The reviewer believes that a brief discussion on organoid applications would be valuable. The introduction could briefly mention the role and potential of organoids in Evolutionary molecular physiology to set the stage for a more detailed discussion in future works. This way, the significance of organoids would be acknowledged while the manuscript's primary focus would be maintained.
A brief introduction to organoids and specific examples of their utility as a model system has been included (Lines 41 - 57) in the most current revision.
Round 3
Reviewer 2 Report
Comments and Suggestions for Authors
It is the reviewer's opinion that after these changes and additions, the manuscript can be accepted for publication.